# A New Approach to the Allocation of Multidimensional Resources in Production Processes

Jarosław Wikarek 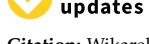 and Paweł Sitek *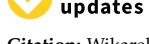

Department of Applied Computer Science, Kielce University of Technology, 25-314 Kielce, Poland; j.wikarek@tu.kielce.pl
* Correspondence: sitek@tu.kielce.pl

**Abstract:** Modern technologies in the field of automation, robotics and IT have significantly changed the face of modern production systems. In particular, the use of AVG, PLC, mobile robots, RFID, IoT, etc. results in modern production processes being characterized by, among others, shortened production cycles and supply chains, reduced production costs, increased product quality and reliability, etc. Moreover, the application of these technologies requires a new definition and methods of using production resources. Most often these are resources that are characterized by many functionalities, the so-called multidimensional resources, which can be configured, remotely controlled, updated, etc., and their use in many cases enables the self-optimization and self-organization of the production system. The article presents the problem of allocation and control of multidimensional resources in production processes. The proprietary formal model of the problem is proposed, as well as how to use it in both proactive and reactive modes. A procedure for reducing the size of the modeled problem is also proposed, the use of which enables a two-fold reduction in the number of constraints and even a fifty-fold reduction in the number of decision variables of the proposed model. This results in an almost hundredfold reduction in computation time for the considered data instances. An original hybrid approach is used to implement the model, which enables the integration of mathematical programming (MP) and programming in constrained logic (CLP). Model data and parameters have been saved as facts.

**Keywords:** resource allocation; multidimensional resources; planning and scheduling; decision support; self-organization/self-optimization systems



## 1. Introduction

The modern economy is characterized by high competitiveness of producers and distributors, global supply chains, fast capital flow, high degree of automation and robotization, etc. On the other hand, an individual approach to the customer and his/her needs is required. All this means that producers and distributors from high-wage countries are under constant pressure to adjust production to the changing and individual customer requirements while minimizing costs, especially labor costs.

The first problem they have to solve is the dichotomy between the scale and scope of production resulting from the diverse needs of customers. Due to the constant increase in labor costs, investments in new technologies and the need to shorten production cycles, it is difficult to be competitive in terms of price and assortment at the same time. This is due to the fact that the economy of scale is based on the production of a large number of products with little diversity, while the economy of scope focuses on providing customized products. Another emerging problem is a clear contradiction between planning and value orientation in production [1]. An answer to the above problems can be the emergence of self-optimization and self-organization production systems. Such systems offer the potential of enhancing flexibility, productivity and reliability of production. The above-mentioned solutions, together with modern technologies such as cyber-physical systems,



RFID, IoT, cloud computing, machine learning, etc., constitute the basis of Industry 4.0 [2,3]. The use of the aforementioned modern technologies in the concept of Industry 4.0 [4] makes it possible to deal with situations of dichotomy related to the quantity and variety of the production assortment. Moreover, it enables adjustment of the configuration and production capacity to the changing external conditions (changing demand, absenteeism, changing distribution conditions, changing legal conditions, etc.).

One of the elements that can be the basis for self-optimization and self-organization production systems [5] is the appropriate approach to the allocation and configuration of modern production resources [6]. This is due to the fact that modern production resources have many functions, can be configured, remotely controlled, etc.

This paper presents the problem of a proactive and reactive approach to the allocation and control of multidimensional resources in production processes.

The structure of the article is as follows. Section 2 contains a literature review on resource allocation problems in production systems. Section 3 presents a description of the problem of multi-dimensional resource allocation under consideration. Section 4 is devoted to the presentation of the proprietary model of multi-dimensional resource allocation, which can be the basis for building a decision support system in the field of modern production systems. Section 5 presents the implementation of the model with the use of two environments, i.e., mathematical programming and constraint logic programming, the so-called hybrid approach [7,8] data structure and an innovative procedure for reducing the size of the modeled problem. Section 6 contains a series of computational experiments that were carried out using multiple data instances differing in the number of workstations, resources, orders, tasks, etc. The last Section is a summary and conclusions.

## 2. Literature Review

Manufacturing resource allocation is an important problem in manufacturing scheduling, which plays a key role in improving manufacturing processes as well as contributes to reducing costs. Manufacturing scheduling problems exist at various levels: individual machine, workstation, shop floor, supply chain, etc. [9]. To simplify, during the scheduling process, we try to make the best possible allocation of orders/tasks to limited resources (or vice versa), taking into account sequence and time constraints. This is a complicated problem of combinatorial optimization known from the literature and practice. It becomes even more difficult if we take into account modern manufacturing systems, such as e.g., smart manufacturing [10].

What makes the presented problem different is, on the one hand, taking into account the multidimensionality of resources (Section 3) and, on the other hand, assuming that the schedule is given/imposed at least in terms of time and sequence constraints. Such a situation is frequent in practice, when the contracting authority requires the execution of the order/project according to its own schedule. However, the key issue remains the allocation of the contractor's resources to tasks/projects.

The presented problem of the allocation of multidimensional resources can be qualified as a certain variant and extension of the generalized assignment problem (GAP). The GAP is defined using the knapsack or the scheduling terminology [11]. Problems related to assignment arise in a number of areas such as transport, education, healthcare, sport, distribution, manufacturing, etc. In practice, this is a well-studied problem in combinatorial optimization with constraints. The GAP refers to research on how to assign n objects (machines, tools, workers, etc.) to m objects (projects, orders, tasks, etc.) in the best possible way (cheapest cost, fastest, etc.) [11,12].

There are two components of the GAP: the assignments and the objective function. In the classical approach, the key question in this problem is: How to carry out an assignment with the optimal objective while at the same time fulfilling all the related constraints? Several methods have been proposed in the literature to answer such a question [12]. The most important ones include exact methods, heuristic methods, population search methods,

and local search methods [11,13–15]. Recently, the Bio-Inspired Hybrid approach seems to be quite promising [16].

What differentiates our approach is the proposal of an original problem model that allows finding answers not to one general question as in the GAP but many, both general and specific questions. Additionally, the model allows for obtaining feedback on what assignments we fail to make and why, which is crucial in the implementation of the project/order. In addition, we also present a proprietary implementation method using the universal AMPL modeling language, a procedure for reducing the size of the modeled problem and two programming environments (MP and CLP).

## 3. Problem Description

The production systems consisting of various manufacturing cells/workstations (w) with any form of production organization (job-shop, flow-shop, open-shop, multi-project, etc.) [17,18] are considered. The single manufacturing cell/workstation [19] can process more than one task/job in parallel. The number of parallel tasks is specified by the SAw parameter. The production system processes a set of tasks/jobs that can be organized as orders or projects (p). The orders/projects implementation schedule is given, which includes the time of their implementation and the sequence of execution. The schedule does not include assignments of jobs/tasks of orders/projects to manufacturing cells/workstations. Additional resources (m) are needed to complete the orders/projects. In the discussed problem, these are the so-called multidimensional resources.

These resources are characterized by the fact that they can have many functionalities (f). Furthermore, these functionalities can be changed or supplemented, which enables the configuration and adjustment of a given resource. Examples of such resources are PLCs with dedicated software, tool magazines with specialized tools, as well as employees with their competencies, etc. On the other hand, similarly to classic resources, they are limited and have an assigned cost of their use, which may depend on the functionality used.

In the systems under consideration, we also take into account the unavailability of certain resources and/or their functionalities. The unavailability of resources is modeled by introducing the parameter (u) defining the so-called state of unavailability. For example, the state u1 = {m1, m3} specifies that the resources m1 and m3 are unavailable, u2 = {m5} that the resource m5 is unavailable, etc.

In the described manufacturing system, three levels can be logically distinguished: the level of additional multidimensional resources, the level of manufacturing cells/workstations and the level of orders/projects. This division is schematically shown in Figure 1.

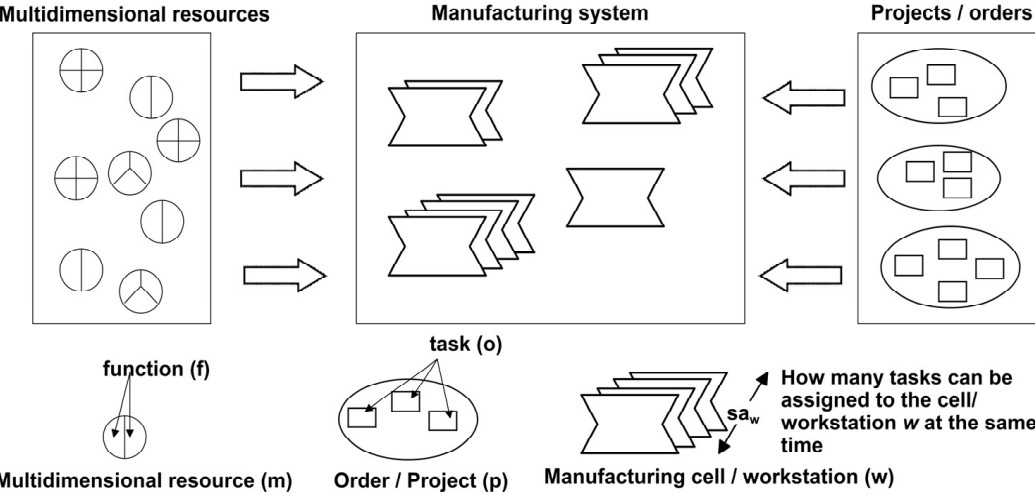

**Figure 1.** Logical levels in the considered production systems.

The use of configurable multidimensional resources in the production system gives the possibility to increase the flexibility of the system under consideration, including through its self-organization and self-optimization. This, in turn, can improve the system's resistance to external factors, reduce production costs, shorten production cycles, etc.

The most important questions that arise in the context of controlling, configuring and allocating multidimensional resources are:

- Are there sufficient multidimensional resources to perform a given set of orders/projects according to a given schedule? (*Q_1*)
- What and how many multidimensional resources and with what functionalities are missing to perform a set of orders/projects according to a given schedule? (*Q_2*)
- Is it possible to complete a set of orders/projects according to a given schedule when the specific multidimensional resource is unavailable? (*Q_3*)
- Which projects should be implemented and how (with what resources) to maximize the profit? (*Q_4*)
- How should a set of multidimensional resources be configured to perform a set of tasks according to a given schedule in the absence of any resource? (*Q_5*)

Finding answers to the above questions is crucial in the context of the implementation of a given set of orders/projects in accordance with the imposed conditions.

As can be seen, individual questions are of different nature. On the one hand, these are general questions (*Q_1*, *Q_3*) and specific questions (*Q_2*, *Q_4*, *Q_5*). On the other hand, they are reactive (*Q_1*, *Q_2*, *Q_4*) or proactive (*Q_5*). Question *Q_3* can be asked in both forms.

Obtaining proactivity is achieved thanks to the appropriate selection and configuration of a set of multidimensional resources, which allows for keeping the production schedule regardless of the lack of availability of the resource or its functionality.

Specifically with respect to question *Q_5*, such a configuration guarantees the execution of a set of orders/projects according to the schedule when some resources or functionalities are unavailable. In this way, a given schedule is resistant to resource unavailability.

## 4. Model of the Allocation and Control of Multidimensional Resources

Experienced production process planners can find answers to questions *Q_1* ... *Q_5* manually. However, this applies to small-scale problems, requires a lot of time, and usually an acceptable solution is found in this way for *Q_4*. In order to automate the method of finding answers to the above questions and to ensure their repeatability, an original model of allocation and control of multidimensional resources has been proposed. The presented model consists of a set of constraints (some of which may be an objective function of the modeled problem) and a set of questions.

The set of indexes and parameters of the model are presented in Table 1. On the other hand, the list of the decision variables of the model and the calculated values is shown in Table 2. The proposed model includes 14 constraints (1) ... (14), the description and interpretation of which is presented in Table 3.

$$\sum_{m \in M} ru_{u,m} \cdot X_{u,m,o,f} = sc_{o,f} - K_{u,o} \forall u \in U, o \in O, f \in F \tag{1}$$

$$N_{m,f} \leq rb_{m,f} \forall m \in M, f \in F \tag{2}$$

$$X_{u,m,o,f} \leq N_{m,f} + ra_{m,f} \forall u \in U, m \in M, o \in O, f \in F \tag{3}$$

$$Count\_1 = \sum_{m \in M} \sum_{f \in F} N_{m,f} \tag{4}$$

$$\sum_{f \in F} X_{u,m,o,f} \leq lb \cdot Y_{u,m,o} \forall u \in U, m \in M, o \in O \tag{5}$$

$$\sum_{f \in F} X_{u,m,o,f} \geq Y_{u,m,o} \forall u \in U, m \in M, o \in O \tag{6}$$

$$\sum_{o2\in O} fo_{o1,o2} \cdot Y_{u,m,o2} \le ar_m \forall u \in U, m \in M, o1 \in O \tag{7}$$

$$Cost\_1 = -\sum_{u\in U}\sum_{m\in M}\sum_{o\in O} (rc_m \cdot ct_o \cdot Y_{u,m,o}) - \sum_{u\in U}\sum_{w\in W}\sum_{o\in O} (ct_o \cdot cr_w \cdot Z_{u,w,o}) + \sum_{u\in U}\sum_{p\in P} (R_{u,p} \cdot vt_p) \tag{8}$$

$$\sum_{m\in M}\sum_{f\in F} rm_{u,w} \cdot Q_{u,w,m,o,f} \le lb \cdot Z_{u,w,o} \forall u \in U, w, \in W, o \in O \tag{9}$$

$$\sum_{w\in W} Z_{u,w,o} \le 1 \forall u \in U, o \in O \tag{10}$$

$$\sum_{o2\in O} fo_{o1,o2} \cdot Z_{u,w,o2} \le sa_w \forall u \in U, w \in W, o1 \in O \tag{11}$$

$$pt_{p,o} \cdot R_{u,p} = pt_{t,o} \cdot K_{u,o} \forall u \in U, p \in P, o \in O \tag{12}$$

$$X_{u,m,o,f} = \sum_{w\in W} (am_{w,m} \cdot rm_{u,w} \cdot Q_{u,w,m,o,f}) \forall u \in U, m \in M, o \in O, f \in F \tag{13}$$

$$
\begin{aligned}
X_{u,m,o,f} &\in \{0,1\} \forall u \in U, m \in M, o \in O, f \in F \\
X1_{u,h,e,z,c} &\in \{0,1\} \forall u \in U, w \in W, m \in M, o \in O, f \in F \\
Z_{u,w,o} &\in \{0,1\} \forall u \in U, w \in W, o \in O \\
Y_{u,m,o} &\in \{0,1\} \forall u \in U, m \in M, o \in O \\
N_{m,f} &\in \{0,1\} \forall m \in M, f \in F \\
K_{u,o} &\in \{0,1\} \forall u \in U, o \in O \\
R_{u,p} &\in \{0,1\} \forall u \in U, p \in P
\end{aligned}
\tag{14}
$$

**Table 1.** Model indexes and parameters.

| Symbol | Description |
|---|---|
| | Indices |
| W | Set of cells/workstations |
| M | Set of multidimensional resources (tools, software, hardware, employees) |
| P | Set of orders/projects |
| O | Set of operations/tasks |
| F | Set of multidimensional resource functionalities/properties |
| U | Set of unavailability states of multidimensional resources and/or cells/workstations |
| m | Multidimensional resource index ($m \in M$) |
| p | Order/project index ($p \in P$) |
| w | Cell/workstation index ($w \in W$) |
| o | Operation/task index ($o \in O$) |
| f | Index of multidimensional resource functionalities/properties ($f \in F$) |
| u | Index of unavailability state ($u \in U$) |
| | Parameters |
| $AR_m$ | Number of available resources m |
| $RC_m$ | Cost of using the resource m per time unit |
| $CT_o$ | Time to complete the task o |
| $SA_w$ | How many tasks can be assigned to the cells/workstations w at the same time |
| $CR_w$ | cell/workstation w work cost per unit of time |
| $VT_p$ | Penalty for failure to complete the order t |

**Table 1.** *Cont.*

| Symbol | Description |
|---|---|
| $PT_{p,o}$ | If task p is part of a project o $PT_{p,o}$ = 1, otherwise $PT_{p,o}$ = 0 |
| $RA_{m,f}$ | If the resource m has the functionality/property f, then $RA_{m,f}$ = 1, otherwise $RA_{m,f}$ = 0 |
| $RB_{m,f}$ | If the resource m can acquire the functionality/property f, then $RB_{m,f}$ = 1, otherwise $RB_{m,f}$ = 0 |
| $SC_{o,f}$ | If a resource functionality/property f is needed to perform the task o, then $SC_{o,f}$ = 1, otherwise $SC_{o,f}$ = 0 |
| $FO_{o1,o2}$ | If the schedule assumes the implementation of task o1, which coincides with the implementation of task o2, then $FO_{o1,o2}$ = 1, otherwise $FO_{o1,o2}$ = 0 |
| $AM_{w,m}$ | If resource m can be allocated to cells/workstations w, $AM_{w,m}$ = 1, otherwise $AM_{w,m}$ = 0 |
| $RU_{u,m}$ | If in the state of unavailability u resource m is available, then $RU_{u,m}$ = 1, otherwise $RU_{u,m}$ = 0 |
| $RM_{u,w}$ | If in the state of unavailability u cells/workstations w is available, then $RM_{u,w}$ = 1, otherwise $RM_{u,w}$ = 0 |
| LB | Arbitrarily large constant |

**Table 2.** Model decision variables and computed values.

| Symbol | Description |
|---|---|
| $X_{u,m,o,f}$ | If in the state of unavailability u resource m performs task o using the functionality/property (competence) f, then $X_{u,m,o,f}$ = 1, otherwise $X_{u,m,o,f}$ = 0 (u ∈ U. m ∈ M, o ∈ O, f ∈ F) |
| $Q_{u,w,m,o,f}$ | If in the state of unavailability u resource m performs task o using the functionality/property f in cells/workstations w, then $Q_{u,w,m,o,f}$ = 1, otherwise $Q_{u,w,m,o,f}$ = 0 (u ∈ U. w ∈ W, m ∈ M, o ∈ O, f ∈ F) |
| $N_{m,f}$ | If the realization sets of tasks require that the resource m can acquire the functionality/property f, then $N_{m,f}$ = 1, otherwise $N_{m,f}$ = 0 (m ∈ M, f ∈ F) |
| $Y_{u,m,o}$ | If in the state of unavailability u resource m performs task o, then $Y_{u,m,o}$ = 1, otherwise $Y_{u,m,o}$ = 0 (u ∈ U, m ∈ M, o ∈ O) |
| $Z_{u,w,o}$ | If in the state of unavailability u task o is performed in the cells/workstations w, $Z_{u,w,o}$ = 1, otherwise $Z_{u,w,o}$ = 0 (u ∈ U, w ∈ W, o ∈ O) |
| $K_{u,o}$ | If in the state of unavailability u due to lack of functionality/property or unavailability of the cells/workstations, the task o cannot be completed, $K_{u,o}$ = 1, otherwise $K_{u,o}$ = 0 (u ∈ U, o ∈ O) |
| $R_{u,p}$ | If in the state of unavailability u project p is performed, $R_{u,p}$ = 0, otherwise $R_{u,p}$ = 1 |
| Cost_1 | Cost of using resources to execute the schedule |
| Count_1 | Number of changes in functionalities/properties needed |

The questions are modeled as subsets of the model constraints (Table 4). Depending on the question being considered, the model may take the form of BILP (Binary Integer Linear Programming) or CSP (Constrain Satisfaction Problem) [20].

**Table 3.** Description of the constraints.

| Constraint | Description |
|---|---|
| (1) | The constraint guarantees that the resources with appropriate functionalities/properties are assigned to each task in any state of unavailability. |
| (2) | The constraint guarantees that the resource concerned can only obtain the functionalities/properties that are permitted for them. |
| (3) | The constraint states that if the selected resource is assigned to a specific task, they must have the required functionalities/properties or can obtain them. |
| (4) | The constraint has defined the number of functionalities/properties changes required |
| (5,6) | Constraints link the decision variables $Y_{u,m,o}$ and $X_{u,m,o,f}$ |
| (7) | The constraint guarantees that simultaneous tasks cannot use the same resources. |
| (8) | The constraint determines the cost of completing the tasks. |
| (9) | Constraints link the decision variables $Q_{u,w,m,o,f}$ and $Z_{u,w,o}$. |
| (10) | The constraint ensures that a given task is performed by only one cell/workstation. |
| (11) | The constraint ensures that the cells/workstations perform only the allowed number of tasks at the same time. |
| (12) | The project/order must be completed in full, i.e., all tasks. |
| (13) | Constraints link the decision variables $Q_{u,w,m,o,f}$ and $X_{u,m,o,f.}$ |
| (14) | Binarity of decision variables |

**Table 4.** Question modelling method.

| Question | Problem | Constrains | Objective | Solution |
|---|---|---|---|---|
| *Q_1* | CSP | (1) . . . (7), (9) . . . (14) Count_1 = 0 | - | $X_{u,m,o,f}$ |
| *Q_2* | BILP | (1) . . . (7), (9) . . . (14) | Min (Count_1) | $N_{m,f}$ |
| *Q_3* | CSP | (1) . . . (7), (9) . . . (14) Count_1 = 0 | - | $X_{w,p,l,c}$ |
| *Q_4* | BILP | (1) . . . (14) | Max (Cost_1) | $X_{w,p,l,c}$ |
| *Q_5* | BILP | (1) . . . (14) | Max (Cost_1) | $X_{w,p,l,c}$ |

## 5. Implementation

AMPL (A Modeling Language for Mathematical Programming) [21] was used to implement the model (*Q_1* . . . *Q_5*, (1) . . . (14)) presented in Section 4. AMPL is a universal tool for modeling problems formulated with the help of mathematical programming. An important advantage of this tool is the ability to work with many solvers such as: BARON, CPLEX, GUROBI, ILOG, LINGO, SCIP, XPRESS, etc. [22], without having to use their modeling languages.

Thus, the problem implementation model in Appendix A can be solved with any of the solvers above. The Gurobi [23] solver was used during the computational experiments presented in Section 6. The data and parameters of the modeled problem were recorded by the facts, the relationship diagram of which is presented in Figure 2 and the description in Table 5. The proposed data representation enables easy integration of decision support models developed in the AMPL language with MRP II, ERP class systems, etc. because the facts are very easy to save in the SQL database.

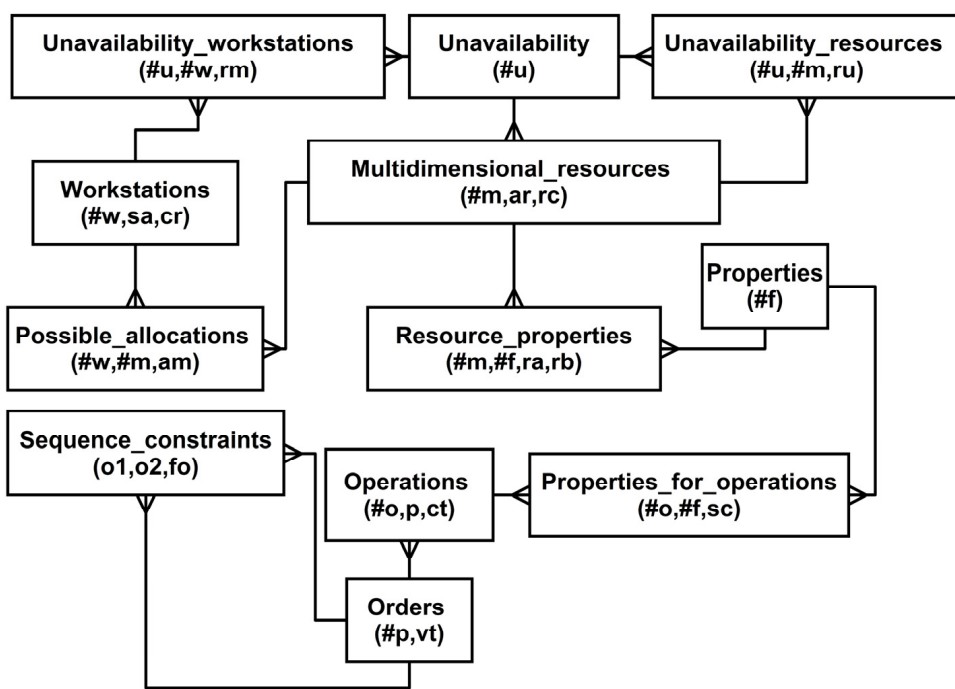

**Figure 2.** Entity Relationship Diagram (ERD) in Martin notation for the database of the problem under consideration.

**Table 5.** Description of individual database entities.

| Entity | Description |
|---|---|
| Multidimensional_resources (#m,ar,rc) | Relationship containing parameters that describe multidimensional resources |
| Workstations (#w,sa,cr) | Relationship containing parameters characterizing cells/workstations |
| Properties (#f) | The relationship describes the individual characteristics of multidimensional resources. |
| Resource_properties (#m,#f,ra,rb) | The relationship determines which resources have which characteristics. |
| Possible_allocations (#w,#m,am) | The relationship specifies possible resource allocations to cells/workstations. |
| Orders (#p,vt) | The report contains a description of projects/orders. |
| Operations (#o,p,ct) | Relationship describing tasks and their belonging to projects |
| Unavailability (#u) | Relationship that identifies the unavailable state |
| Unavailability_resources (#u,#m,ru) | Relationship characterizing the unavailability of resources |
| Unavailability_workstations (#u,#w,rm) | Relationship characterizing the unavailability of cells/workstations |
| Sequence_constraints (o1,o2,fo) | Task sequence constraints relationship |
| Properties_for_operations (#o,#f,sc) | Relationship defining what features are needed to accomplish particular tasks |

Due to its combinatorial nature as well as binary decision variables, the modeled problem is NP-Hard. Therefore, for problems of larger sizes, approximate methods should be used to solve them, e.g., using artificial intelligence techniques, dedicated heuristics, etc.

For such cases, the authors used an innovative approach that allows the size of the modeled problem to be reduced, which results in a reduction of the solution space and, consequently, shortens the computation time. A proprietary procedure has been proposed to reduce the size of the problem based on facts (Figure 3). The fact notation contains only non-zero values of individual parameters of the modeled problem, while solvers require matrix ones, which results in the formation of large-sized matrices containing many zeros/sparse matrices.

Step 1. For every constraint.
    Step 2. For each fact that describes the problem.
        Step 3. If the fact keys are a subset of the constraint attributes.
            Step 4. If there is no fact for the attribute values.
            Step 5. Remove this restriction.
            Step 6. Go to the next restriction.
        Step 7. Otherwise.
            Step 8. For each variable in this constraint.
                Step 9. If the attributes of the relationship are a subset of the variable's attributes.
                Step 10. If there is no fact about the attribute values of the variable.
                Step 11 Remove this variable from the constraint.

**Figure 3.** Procedure for reducing the size of the modeled problem (pseudocode).

The proposed procedure (Figure 3), to a large extent, works as follows. Based on the value of the facts, it determines which coefficients/model parameters are equal to zero/are not present in the facts. On this basis and using the model structure, it sets the zero values of the decision variables present at these parameters. In this way, the number of decision variables is reduced and the structure of some constraints, etc., is simplified.

Some kind of presolving the modeled problem is performed with this procedure. The procedure itself exhibits a polynomial computational complexity. After applying the procedure, a reduced model is obtained, which has the same nature as before its application. Therefore, it can be solved with any solver. Moreover, the proposed method gives exact solutions. The facts (Appendix B) were saved in a relational database, the diagram of which is presented in Figure 2. The key attributes of individual entities were preceded by the # sign. The description of individual database entities is presented in Table 5. Exactly one row of the selected table corresponds to this fact. This type of recording facilitates integration with MRP II (Manufacturing Resource Planning), ERP (Enterprise Resources Planning), etc. systems.

## 6. Illustrative Example and Computational Experiments

In an illustrative example, a production system is given (Figure 4) which consists of 5 cells/workstations (w = w01 . . . w05). The individual cell/workstation can process from two to four tasks simultaneously ($SA_w$) and has a specific unit production cost ($CR_w$). There are 14 additional resources in the system (m = m01 . . . m14), which can have up to 12 functionalities (f = f01 . . . f12) and six projects/orders (p = p01 . . . p06) consisting of 20 tasks (o = o01 . . . o20) that should be completed in the system. There is also a projects/orders execution schedule that defines the time and order of execution of tasks from the orders (Figure 5). The schedule does not define assignments of tasks or resources to cells/workstations. Individual projects are marked with different colors according to the principle as in Figure 5.

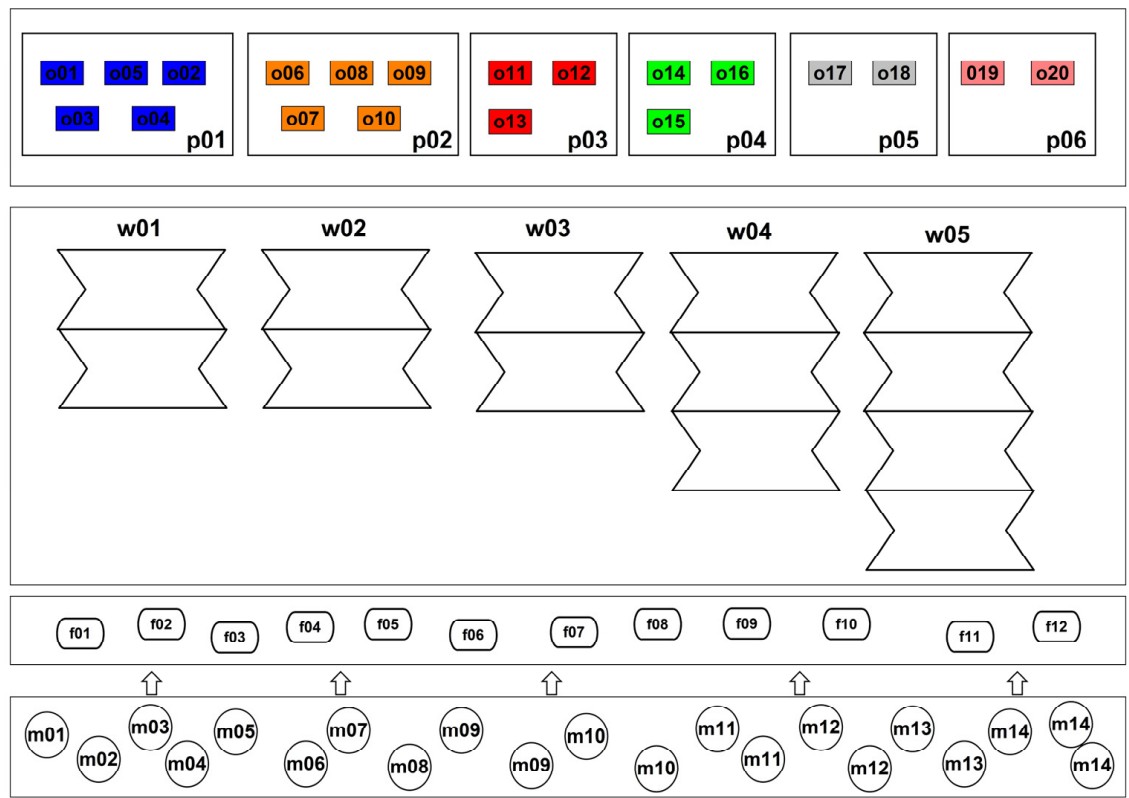

**Figure 4.** The layout of production system for illustrative example.

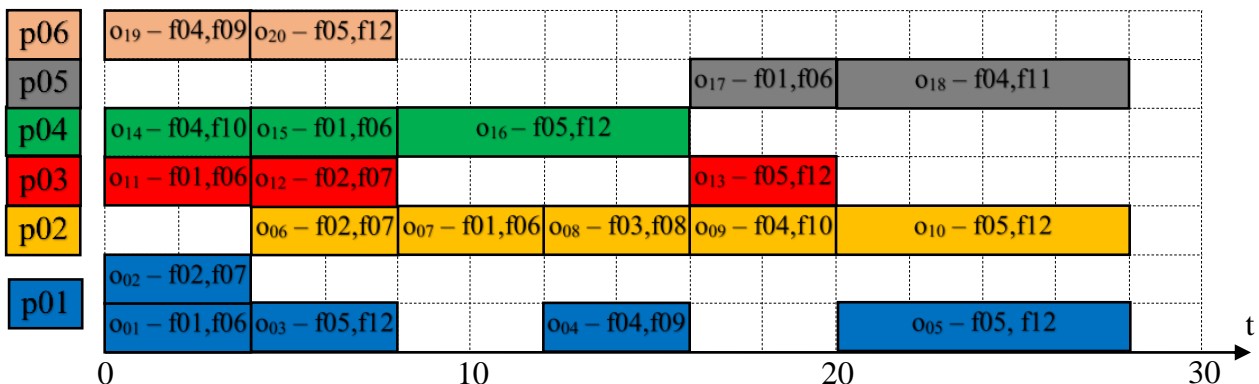

**Figure 5.** Execution schedule for illustrative example.

All calculation examples were carried out using a computer with the following parameters: Intel(R) Core(TM) i5-4200 M processor, CPU @ 2.50 GHz, 800 GB RAM.

Computational experiments with the use of the proposed model (Section 4) were carried out in two stages. In the first stage, the answers to questions *Q_1 … Q_5* were searched for the illustrative example data instance (Appendix A). During the computational experiments, the AMPL model and the GUROBI solver were used.

The first answer was to question *Q_1* (Are there sufficient additional resources to perform a given set of tasks according to a given schedule?), which was YES. The responding cost (Cost_1) of completing the tasks was 238,000. The corresponding system configuration (which means the allocation of multidimensional resources with specific features to the appropriate workstations) at a selected point in time ($\tau = 2$) is presented in Figure 6, whereas the detailed implementation schedule with multidimensional resources and their features allocated to tasks is presented in Figure 7.

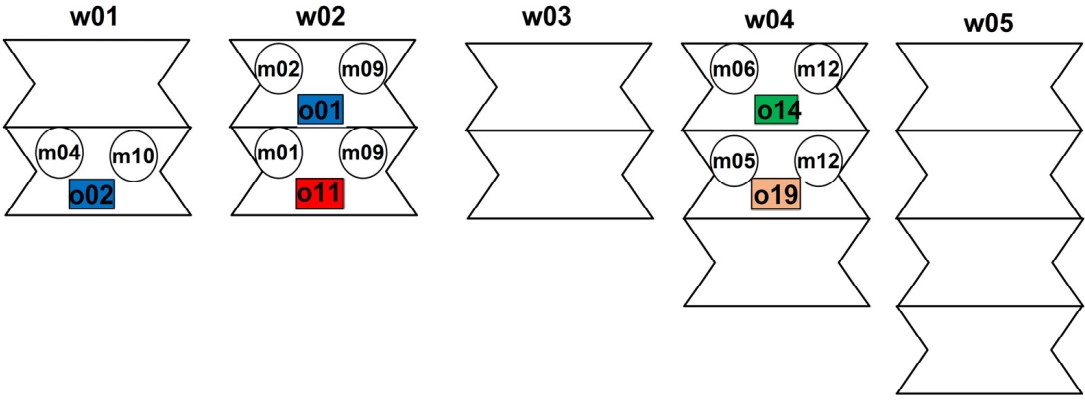

**Figure 6.** System configuration for a moment of time τ = 2 (concerns question *Q_1*).

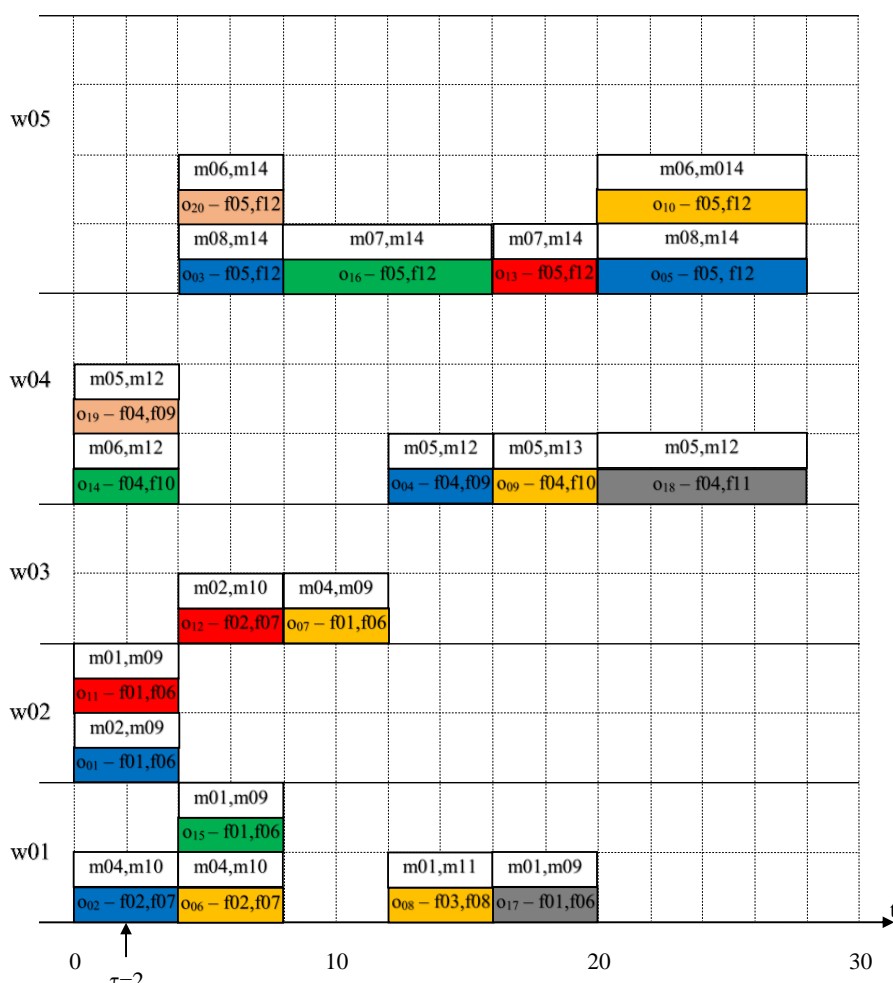

**Figure 7.** Project implementation schedule with the assignment of tasks and multidimensional resources to workstations for *Q_1* (individual projects have different colors as shown in Figure 5).

The answer to question *Q_1* (What and how many additional resources and with what qualities are missing to perform a set of tasks according to a given schedule?) was 'not missing any resources or functionality', see *Q_1*.

Meanwhile, the answer to question *Q_3* (Is it possible to perform a set of tasks according to a given schedule when the specific additional resource is unavailable?) was considered for each resource. The answers are presented in Table 6.

**Table 6.** Answers for all variants of question *Q_3*.

| Resource Unavailable | Answer |
| --- | --- |
| m01, m02, m03, m04, m05, m06, m07, m08, m13 | YES, accordingly the Cost_1 244,520, 244,120, 244,120, 243,320, 244,520, 243,320, 244,520, 243,000, 244,360 |
| m09 | NO, two f06 functionalities are missing. |
| m10 | NO, two f07 functionalities are missing. |
| m11 | NO, f08 functionality is missing. |
| m12 | NO, f11 functionality is missing. |
| m14 | NO, f12 functionality is missing. |

The answer to question *Q_4* (What is the minimum cost of resource allocation to perform a set of tasks according to a given schedule?) is as follows: the cost (Cost_1) of completing the tasks was 244,520. The corresponding system configuration at a selected point in time ($\tau = 2$) is presented in Figure 8, while the detailed implementation schedule with allocated multidimensional resources and features for tasks is presented in Figure 9.

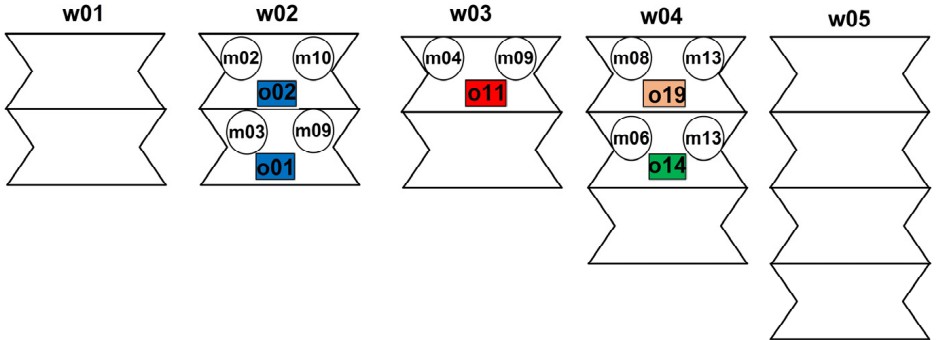

**Figure 8.** System configuration for a moment of time $\tau = 2$ (concerns the question *Q_4*).

Whereas the answer to question *Q_5* is the aggregation of all answers to question *Q_3*. Such a supplemented configuration with missing resource functionalities (Table 6) enables the implementation of a set of tasks according to the given schedule when any of the resources m1 . . . m14 is unavailable.

The obtained results (i.e., answers to questions *Q_1* . . . *Q_5*) allow:

- finding the production system configuration at a given time, i.e., which tasks will be performed on which machines/workstations with the use of what resources (Figures 6–9);
- defining the configuration of the resources themselves, i.e., what features/functions of the resource are used;
- in the case of impossibility of implementation, it will be asked to indicate what resources and/or features are missing (Table 6);
- proactively predict whether the set of tasks will be carried out in the absence of a resource/feature.

In the second stage of the experiments, the effectiveness of the applied methods of implementing the model was examined. The research was carried out for a larger number of instances of larger data. Individual data instances differed in the number of workstations, projects, tasks, etc. For the experiments, the variants of models with questions *Q_4* and *Q_5* were selected because they required the greatest computational outlays. Two implementation methods were used during the experiments. The first is analogous to the illustrative example with the use of the AMPL and Gurobi solvers. The second method involved the use of the presolving procedure. This procedure (Figure 3) makes it possible to reduce the size of the modeled problem without changing its nature.

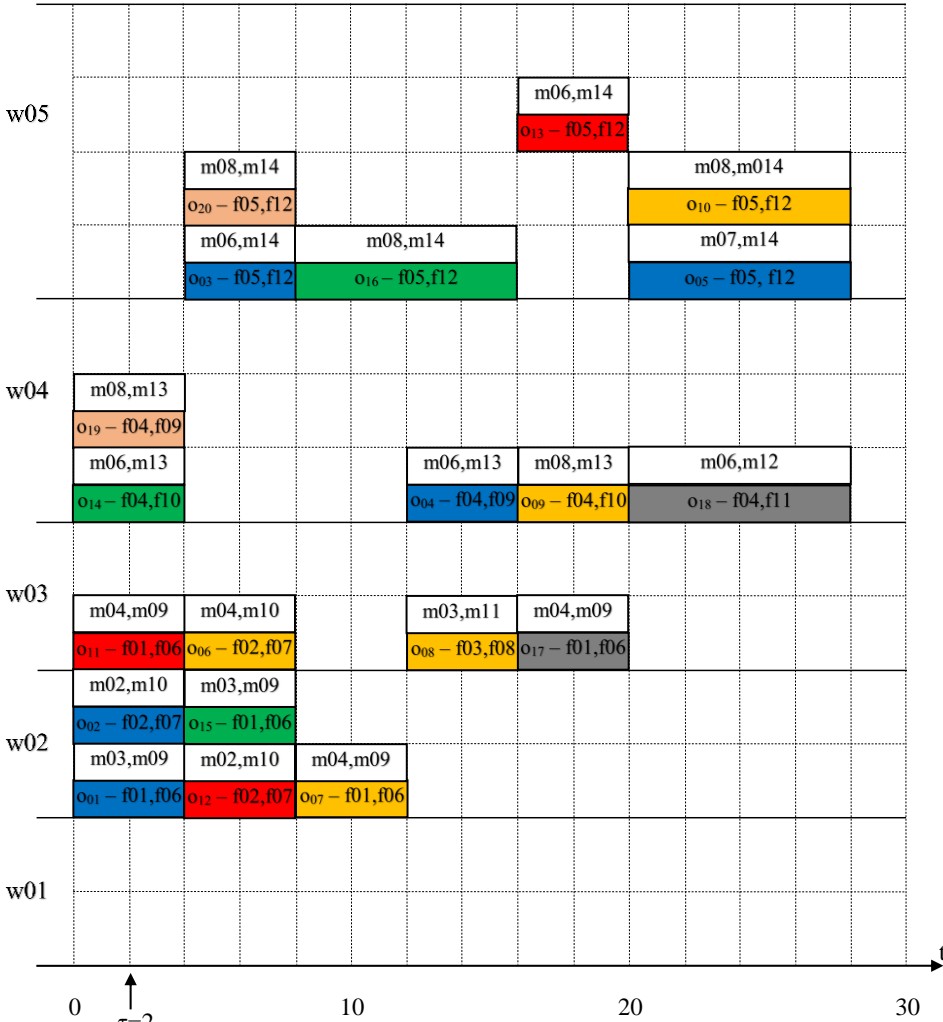

**Figure 9.** Project implementation schedule with the assignment of tasks and multidimensional resources to workstations for *Q_4* (individual projects have different colors as shown in Figure 5).

The obtained results are presented in Table 7. The first five lines refer to question *Q_4* and the remaining lines to *Q_5*. As it is easy to notice when analyzing the obtained results, the application of the presolving procedure during the implementation resulted in a significant reduction in the size of the modeled problems. Thus, depending on the data instance and question, the number of decision variables of the modeled problem was reduced by 30 to 50 times, while the number of constraints was reduced by about two times.

However, the most important practical effect of using this procedure was the reduction of computation time. It was surprisingly high and amounted to even several hundred times. Moreover, for some data instances, it was not possible to find a solution within the acceptable time (1200 s) in the implementation without the use of a procedure. After its application, finding the optimal solution took a few seconds. Another aspect of the use of the procedure is the possibility of solving practical problems of real/industrial/sizes using exact methods (any mathematical solvers).

**Table 7.** Answers for *Q4* (1–5) and *Q5* (6–10) for different data instances.

| N | The Dimensions of the Problem | | | | | | Without Presolving | | | With Presolving | | |
|---|---|---|---|---|---|---|---|---|---|---|---|---|
| | | | Number of Facts | | | | Number of Decision Variables | Number of Constrains | Computation Time (s) | Number of Decision Variables | Number of Constrains | Computation Time (s) |
| | Unavailability | Workstations | Multidimensional_Resources | Tasks | Project | Properties | | | | | | |
| | u | h | e | z | c | t | | | | | | |
| 1 | 1 | 5 | 14 | 20 | 12 | 6 | 20,734 | 10,690 | 4 | 796 | 5303 | 1 |
| 2 | 1 | 8 | 16 | 24 | 12 | 6 | 42,270 | 14,834 | 34 | 916 | 6951 | 1 |
| 3 | 1 | 8 | 16 | 28 | 14 | 6 | 57,378 | 19,826 | 156 | 1319 | 9151 | 1 |
| 4 | 1 | 10 | 20 | 34 | 18 | 8 | 136,062 | 37,694 | 458 | 3238 | 17,068 | 2 |
| 5 | 1 | 10 | 20 | 38 | 18 | 10 | 152,028 | 46,266 | 1200 * | 3549 | 19,121 | 2 |
| 6 | 5 | 5 | 14 | 20 | 12 | 6 | 102,998 | 43,170 | 589 | 3138 | 23,445 | 1 |
| 7 | 5 | 8 | 16 | 24 | 12 | 6 | 210,582 | 59,570 | 768 | 4589 | 32,454 | 1 |
| 8 | 5 | 8 | 16 | 28 | 14 | 6 | 285,994 | 79,410 | 1200 * | 6890 | 43,145 | 2 |
| 9 | 5 | 10 | 20 | 34 | 18 | 8 | 678,870 | 150,030 | 1200 * | 13570 | 79,633 | 7 |
| 10 | 5 | 10 | 20 | 38 | 18 | 10 | 758,700 | 172,122 | 1200 ** | 15,204 | 88,750 | 12 |

* Only feasible solution; ** No solution found.

### 7. Conclusions

The proposed model of the allocation and control of multidimensional resources can be the basis for supporting decisions in the production system in terms of selecting and controlling resources, optimizing production costs, accepting or rejecting a new project, etc. Additionally, it enables finding optimal/acceptable allocations of multidimensional resources to individual project/order tasks. Moreover, the model can be used to support both proactive and reactive decisions. Due to the model structure, which consists of constraints and the set of questions, the model is very easy to modify and extend by adding/reducing constraints and questions. Saving the data and parameters of the model in a relational database facilitates its integration with MRPII, ERP, etc. systems [24].

An important contribution is the implementation of the model using the AMPL solver. This makes it possible to solve the modeled problem using most of the solvers available on the market. However, a key element of the proposed implementation is the procedure for reducing the size of the modeled problem (Figure 3). This procedure de facto enables presolving, as a result of which we obtain a significant reduction in the number of decision variables and model constraints (Table 7). The proposed procedure does not change the nature of the model; therefore, after its application, we can still solve it using a mathematical programming/constraint programming solver, obtaining an exact solution in significantly less time (even a hundred times less).

In further research, we plan to:

- Introduce new questions such as: Is it possible to implement a new project/order with a given schedule with the current multidimensional resources and cells/workstations? for other variants of production, logistics and distribution systems [25–29].
- Use the presented procedure in conjunction with metaheuristic methods such as genetic algorithms, bio-inspired algorithms [30] as well hybrid methods [31,32];
- Expand the proposed model with the functionality of creating an acceptable schedule in the event of impossible implementation with the resources at hand of a given schedule.

**Author Contributions:** Conceptualization, P.S.; methodology, P.S. and J.W.; software, J.W.; validation, P.S. and J.W.; formal analysis, J.W.; investigation, P.S. and J.W.; resources, J.W.; data curation, J.W.; writing—original draft preparation, P.S..; writing—review and editing, J.W.; visualization J.W.; supervision P.S.; project administration, P.S.; funding acquisition, J.W. All authors have read and agreed to the published version of the manuscript.

**Funding:** This research received no external funding.

**Institutional Review Board Statement:** Not applicable.

**Informed Consent Statement:** Not applicable.

**Data Availability Statement:** Not applicable.

**Conflicts of Interest:** The authors declare no conflict of interest.

### Appendix A. AMPL Model of Allocation and Control of Multidimensional Resources

```
set WS;  set MS;  set PS;
set OS;  set FS;  set US;
param   sa\{WS\};   param   cr\{WS\};
param   ar\{MS\};   param   rc\{MS\};
param   vt\{PS\};   param   ct\{OS\};
param   rb\{MS,FS\};    param   ra\{MS,FS\};
param   sc\{OS,FS\};    param   fo\{OS,OS\};
param   am\{WS,MS\};    param   ru\{US,MS\};
param   rm\{US,WS\};    param   pt\{PS,OS\};
param   lb;
var   K\{US,OS\}   \textgreater{}=0, binary;
```

```
var    N\{MS,FS\}    \textgreater{}=0, binary;
var    X\{US,MS,OS,FS\}    \textgreater{}=0, binary;
var    Y\{US,MS,OS\}    \textgreater{}=0, binary;
var    Z\{US,WS,OS\}    \textgreater{}=0, binary;
var    Q\{US,WS,MS,OS,FS\} \textgreater{}=0, binary;
var    R\{US,PS\}    \textgreater{}=0, binary;
var    Count\_1; var    Cost\_1;
subject to O1 \{u in US, o in OS, f in FS :
  sc[o,f]=1\}:sum\{m in MS\} X[u,m,o,f]=
  sc[o,f]*K[u,o];
subject to O2 \{m in MS, f in FS \}:
  N[m,f]\textless{}=rb[m,f];
subject to O3 \{u in US, m in MS, o in OS,
  f in FS\}:X[u,m,o,f]\textless{}=N[m,f]+ra[m,f];
subject to O4 :
  Count\_1=sum\{m in MS, f in FS\} N[m,f];
subject to O5 \{u in US, m in MS, o in OS\}:
  sum\{f in FS\} X[u,m,o,f]\textless{}=lb*Y[u,m,o];
subject to O6 \{u in US, m in MS, o in OS\}:
  sum\{f in FS\} X[u,m,o,f]\textgreater{}=Y[u,m,o];
subject to O7 \{u in US, m in MS, o1 in OS\}:
  sum\{o2 in OS\} fo[o1,o2]*Y[u,m,o2]\textless{}=ar[m];
subject to O8 : Cost\_1=-sum\{u in US, m in MS,
  o in OS\} Y[u,m,o]*rc[m]*ct[o] -
  sum\{u in US, w in WS, o in OS\}
  ct[o]*cr[w]*Z[u,w,o]+
  sum\{u in US, p in PS\} R[u,p]*vt[p];
subject to O9 \{u in US, w in WS, o in OS \}:
  sum\{m in MS, f in FS\} rm[u,w]*Q[u,w,m,o,f]\textless{}=
  lb*Z[u,w,o];
subject to O10 \{u in US, o in OS\}:
  sum\{w in WS\} Z[u,w,o] \textless{}=1;
subject to O11 \{u in US, w in WS,
  o1 in OS \}: sum\{o2 in OS\}
  fo[o1,o2]*Z[u,w,o2]\textless{}=sa[w];
subject to O12 \{u in US, p in PS, o in OS\}:
  pt[p,o]*R[u,p]= pt[p,o]*K[u,o];
subject to O13 \{u in US, m in MS, o in OS,
  f in FS \}:X[u,m,o,f]=sum\{w in WS\}
  am[w,m]*rm[u,w]*Q[u,w,m,o,f];

data;
set WS :=  w01 w02 w03 w04 w05 ;
set MS :=  m01 m02 m03 m04 m05 m06 m07 m08
  m09 m10 m11 m12 m13 m14;
set FS :=  f01 f02 f03 f04 f05 f06 f07 f08
  f09 f10 f11 f12;
set PS :=  p01 p02 p03 p04 p05 p06 ;
set OS :=  o01 o02 o03 o04 o05 o06 o07 o08
   o09 o10 o11 o12 o13 o14 o15 o16 o17 o18
   o19 o20;
set US :=  u01;
param vt := p01 20000 p02 18000 p03 80000 p04 80000 p05 50000 p06 50000;
param ar := m01 1 m02 1 m03 1 m04 1 m05 1 m06 1 m07 1 m08 1 m09 2 m10 2
```

```
      m11 2 m12 2 m13 2 m14 3 ;
param rc := m01 200 m02 150 m03 150 m04 100 m05 200 m06 150 m07 200 m08
    140 m09 60 m10 50 m11 50 m12 90 m13 80 m14 40;
param ct := o01 4 o02 4 o03 4 o04 4 o05 8 o06 4 o07 4 o08 4 o09 4 o10 8
    o11 4 o12 4 o13 4 o14 4 o15 4 o16 8 o17 4 o18 8 o19 4 o20 8;
param sa := w01 1 w02 1 w03 1 w04 3 w05 4;
param cr:= w01 600 w02 400 w03 400 w04 300 w05 300;
param pt : o01 o02 o03 o04 o05 o06 o07 o08 o09 o10 o11 o12 o13 o14 o15
    o16 o17 o18 o19 o20 := p01  1 1 1 1 1 0 0 0 0 0 0 0 0 0 0 0 0 0 0
    0 p02  0 0 0 0 0 1 1 1 1 1 0 0 0 0 0 0 0 0 0 0 p03  0 0 0 0 0 0 0
    0 0 0 1 1 1 0 0 0 0 0 0 0 p04  0 0 0 0 0 0 0 0 0 0 0 0 0 1 1 1 0 0
    0 0 p05  0 0 0 0 0 0 0 0 0 0 0 0 0 0 0 0 1 1 0 0 p06 0 0 0 0 0 0 0
    0 0 0 0 0 0 0 0 0 0 0 1 1;
param ra: f01 f02 f03 f04 f05 f06 f07 f08 f09 f10 f11 f12 :=  m01 1 1 1 1
    0 0 0 0 0 0 0 0 m02 1 1 1 1 0 0 0 0 0 0 0 0  m03 1 1 1 1 0 0 0 0 0
    0 0 m04 1 1 1 1 0 0 0 0 0 0 0 0 m05 0 0 0 1 0 0 0 0 0 0 0 0 m06
    0 0 0 1 1 0 0 0 0 0 0 0 m07 0 0 0 0 1 0 0 0 0 0 0 0 m08 0 0 0 1 1
    0 0 0 0 0 0 0 m09 0 0 0 0 0 1 0 0 0 0 0 0 m10 0 0 0 0 0 0 1 0 0 0
    0 0 m11 0 0 0 0 0 0 0 1 0 0 0 0 m12 0 0 0 0 0 0 0 0 1 1 1 0  m13 0
    0 0 0 0 0 0 0 1 1 0 0 m14 0 0 0 0 0 0 0 0 0 0 0 1;
param rb: f01 f02 f03 f04 f05 f06 f07 f08 f09 f10 f11 f12 := m01 0 0 0 0
    0 0 0 0 0 0 0 0 m02 0 0 0 0 0 0 0 0 0 0 0 0 m03 0 0 0 0 0 0 0 0 0
    0 0 0 m04 0 0 0 0 0 0 0 0 0 0 0 0 m05 0 0 0 0 0 0 0 0 0 0 0 0 m06
    0 0 0 0 0 0 0 0 0 0 0 0 m07 0 0 0 0 0 0 0 0 0 0 0 0 m08 0 0 0 0 0
    0 0 0 0 0 0 0 m09 0 0 0 0 0 0 0 0 0 0 0 0 m10 0 0 0 0 0 0 0 0 0 0
    0 0 m11 0 0 0 0 0 0 0 0 0 0 0 0 m12 0 0 0 0 0 0 0 0 0 0 0 0 m13 0
    0 0 0 0 0 0 0 0 0 0 0 m14 0 0 0 0 0 0 0 0 0 0 0 0;
param sc: f01 f02 f03 f04 f05 f06 f07 f08 f09 f10 f11 f12 := o01 1 0 0 0
    0 1 0 0 0 0 0 0 o02 0 1 0 0 0 0 1 0 0 0 0 0 o03 0 0 0 0 1 0 0 0 0
    0 0 1 o04 0 0 0 1 0 0 0 0 1 0 0 0 o05 0 0 0 0 1 0 0 0 0 0 0 1 o06
    0 1 0 0 0 0 1 0 0 0 0 0 o07 1 0 0 0 0 1 0 0 0 0 0 0 o08 0 0 1 0 0
    0 0 1 0 0 0 0 o09 0 0 0 1 0 0 0 0 1 0 0 o10 0 0 0 0 1 0 0 0 0 0 0
    0 1 o11 1 0 0 0 0 1 0 0 0 0 0 0 o12 0 1 0 0 0 0 1 0 0 0 0 0 o13 0
    0 0 0 1 0 0 0 0 0 1 o14 0 0 0 1 0 0 0 0 1 0 0 o15 1 0 0 0 0 1
    0 0 0 0 0 0 o16 0 0 0 0 1 0 0 0 0 0 1 o17 1 0 0 0 0 1 0 0 0 0 0 0
    0 o18 0 0 0 1 0 0 0 0 0 0 1 0 o19 0 0 0 1 0 0 0 0 1 0 0 0 o20 0 0
    0 0 1 0 0 0 0 0 0 1;
param fo: o01 o02 o03 o04 o05 o06 o07 o08 o09 o10 o11 o12 o13 o14 o15 o16
    o17 o18 o19 o20 := o01 0 1 0 0 0 0 0 0 0 0 1 0 0 1 0 0 0 0 1 0 o02
    1 0 0 0 0 0 0 0 0 0 1 0 0 1 0 0 0 0 1 0 o03 0 0 0 0 0 1 0 0 0 0 0
    1 0 0 1 0 0 0 0 1 o04 0 0 0 0 0 0 1 0 0 0 0 0 1 0 0 0 0 o05
    0 0 0 0 0 0 0 0 1 0 0 0 0 0 1 0 0 o06 0 0 1 0 0 0 0 0 0 0
    1 0 0 1 0 0 0 0 1 o07 0 0 0 0 0 0 0 0 0 0 0 0 0 1 0 0 0 0 o08
    0 0 0 1 0 0 0 0 0 0 0 0 0 1 0 0 0 0 o09 0 0 0 0 0 0 0 0 0 0 0
    0 1 0 0 0 1 0 0 0 o10 0 0 0 1 0 0 0 0 0 0 0 0 0 0 0 1 0 0 o11
    1 1 0 0 0 0 0 0 0 0 0 1 0 0 0 0 1 0 o12 0 0 1 0 0 1 0 0 0 0 0
    0 0 0 1 0 0 0 0 1 o13 0 0 0 0 0 0 0 0 1 0 0 0 0 0 0 0 1 0 0 0 o14
    1 1 0 0 0 0 0 0 0 0 0 1 0 0 0 0 0 0 1 0 o15 0 0 1 0 0 1 0 0 0 0 0
    0 0 0 1 0 0 0 0 1 o16 0 0 0 1 0 0 1 1 0 0 0 0 0 0 0 0 0 0 0 0 o17
    0 0 0 0 0 0 0 0 1 0 0 0 1 0 0 0 0 0 0 o18 0 0 0 0 1 0 0 0 0 1 0
    0 0 0 0 0 0 0 o19 1 1 0 0 0 0 0 0 0 0 1 0 0 1 0 0 0 0 0 0 o20
    0 0 1 0 0 1 0 0 0 0 0 1 0 0 0 0 0 0 0 1;
param am: m01 m02 m03 m04 m05 m06 m07 m08 m09 m10 m11 m12 m13 m14 := w01
    1 1 1 1 0 0 0 0 1 1 1 0 0 0 w02 1 1 1 1 0 0 0 0 1 1 1 0 0 0 w03 1
```

```
            1 1 1 0 0 0 0 1 1 1 0 0 0 w04 0 0 0 0 1 1 1 1 0 0 1 1 1 0 w05 0 0
            0 0 1 1 1 1 0 0 0 0 0 1;
param ru: m01 m02 m03 m04 m05 m06 m07 m08 m09 m10 m11 m12 m13 m14 :=  u01
        1 1 1 1 1 1 1 1 1 1 1 1 1 1;
param rm: w01 w02 w03 w04 w05 :=
 u01 1 1 1 1 1  ;
param lb:= 1000000;
```

## Appendix B. Data and Parameters of the Problem in the Form of Facts

```
%orders(p,vt),
orders(p1,20000). orders(p2,18000). orders(p3,80000).
orders(p4,8000).  orders(p5,50000). orders(p6,50000).
%operations(o,p,ct),
operations(o1,p1,4).  operations(o2,p1,4).  operations(o3,p1,4).
operations(o4,p1,4).  operations(o5,p1,8).  operations(o6,p2,4).
operations(o7,p2,4).  operations(o8,p2,4).  operations(o9,p2,4).
operations(o10,p2,8). operations(o11,p3,4). operations(o12,p3,4).
operations(o13,p3,4). operations(o14,p4,4). operations(o15,p4,4).
operations(o16,p4,8). operations(o17,p5,4). operations(o18,p5,8).
operations(o19,p6,4). operations(o20,p6,8).
%unavailability(u),
unavailability(u1).
%multidimensional_resources(m,ar,rc),
multidimensional\_resources(m1,1,200). multidimensional\_resources(m2,1,150).
multidimensional\_resources(m3,1,150). multidimensional\_resources(m4,1,100).
multidimensional\_resources(m5,1,200). multidimensional\_resources(m6,1,150).
multidimensional\_resources(m7,1,200). multidimensional\_resources(m8,1,140).
multidimensional\_resources(m9,2,60).  multidimensional\_resources(m10,2,50).
multidimensional\_resources(m11,2,50). multidimensional\_resources(m12,2,90).
multidimensional\_resources(m13,2,80). multidimensional\_resources(m14,3,40).
%properties(f),
properties(f1). properties(f2).  properties(f3).  properties(f4).
properties(f5). properties(f6).  properties(f7).  properties(f8).
properties(f9). properties(f10). properties(f11). properties(f12).
%properties_for_operations(o,f,sc),
properties\_for\_operations(o1,f1,1).  properties\_for\_operations(o1,f6,1).
properties\_for\_operations(o2,f2,1).  properties\_for\_operations(o2,f7,1).
properties\_for\_operations(o3,f5,1).  properties\_for\_operations(o3,f12,1).
properties\_for\_operations(o4,f4,1).  properties\_for\_operations(o4,f9,1).
properties\_for\_operations(o5,f5,1).  properties\_for\_operations(o5,f12,1).
properties\_for\_operations(o6,f2,1).  properties\_for\_operations(o6,f7,1).
properties\_for\_operations(o7,f1,1).  properties\_for\_operations(o7,f6,1).
properties\_for\_operations(o8,f3,1).  properties\_for\_operations(o8,f8,1).
properties\_for\_operations(o9,f4,1).  properties\_for\_operations(o9,f10,1).
properties\_for\_operations(o10,f5,1). properties\_for\_operations(o10,f12,1).
properties\_for\_operations(o11,f1,1). properties\_for\_operations(o11,f6,1).
properties\_for\_operations(o12,f2,1). properties\_for\_operations(o12,f7,1).
properties\_for\_operations(o13,f5,1). properties\_for\_operations(o13,f12,1).
properties\_for\_operations(o14,f4,1). properties\_for\_operations(o14,f10,1).
properties\_for\_operations(o15,f1,1). properties\_for\_operations(o15,f6,1).
properties\_for\_operations(o16,f5,1). properties\_for\_operations(o16,f12,1).
properties\_for\_operations(o17,f1,1). properties\_for\_operations(o17,f6,1).
properties\_for\_operations(o18,f4,1). properties\_for\_operations(o18,f11,1).
properties\_for\_operations(o19,f4,1). properties\_for\_operations(o19,f9,1).
properties\_for\_operations(o20,f5,1). properties\_for\_operations(o20,f12,1).
```

```
%workstations(w,sa,cr),
workstations(w1,2,600). workstations(w2,2,400).
workstations(w3,2,400). workstations(w4,3,300).
workstations(w5,4,300).
%resource_properties(m,f,ra,rb),
resource\_properties(m1,f1,1,0).    resource\_properties(m1,f2,1,0).
resource\_properties(m1,f3,1,0).    resource\_properties(m1,f4,1,0).
resource\_properties(m2,f1,1,0).    resource\_properties(m2,f2,1,0).
resource\_properties(m2,f3,1,0).    resource\_properties(m2,f4,1,0).
resource\_properties(m3,f1,1,0).    resource\_properties(m3,f2,1,0).
resource\_properties(m3,f3,1,0).    resource\_properties(m3,f4,1,0).
resource\_properties(m4,f1,1,0).    resource\_properties(m4,f2,1,0).
resource\_properties(m4,f3,1,0).    resource\_properties(m4,f4,1,0).
resource\_properties(m5,f4,1,0).    resource\_properties(m6,f4,1,0).
resource\_properties(m6,f5,1,0).    resource\_properties(m7,f5,1,0).
resource\_properties(m8,f4,1,0).    resource\_properties(m8,f5,1,0).
resource\_properties(m9,f6,1,0).    resource\_properties(m10,f7,1,0).
resource\_properties(m11,f8,1,0).   resource\_properties(m12,f9,1,0).
resource\_properties(m12,f10,1,0).  resource\_properties(m12,f11,1,0).
resource\_properties(m13,f9,1,0).   resource\_properties(m13,f10,1,0).
resource\_properties(m14,f12,1,0).
%possible_allocations(w,m,am),
possible\_allocations(w1,m1,1).  possible\_allocations(w1,m2,1).
possible\_allocations(w1,m3,1).  possible\_allocations(w1,m4,1).
possible\_allocations(w1,m9,1).  possible\_allocations(w1,m10,1).
possible\_allocations(w1,m11,1). possible\_allocations(w2,m1,1).
possible\_allocations(w2,m2,1).  possible\_allocations(w2,m3,1).
possible\_allocations(w2,m4,1).  possible\_allocations(w2,m9,1).
possible\_allocations(w2,m10,1). possible\_allocations(w2,m11,1).
possible\_allocations(w3,m1,1).  possible\_allocations(w3,m2,1).
possible\_allocations(w3,m3,1).  possible\_allocations(w3,m4,1).
possible\_allocations(w3,m9,1).  possible\_allocations(w3,m10,1).
possible\_allocations(w3,m11,1). possible\_allocations(w4,m5,1).
possible\_allocations(w4,m6,1).  possible\_allocations(w4,m7,1).
possible\_allocations(w4,m8,1).  possible\_allocations(w4,m11,1).
possible\_allocations(w4,m12,1). possible\_allocations(w4,m13,1).
possible\_allocations(w5,m5,1).  possible\_allocations(w5,m6,1).
possible\_allocations(w5,m7,1).  possible\_allocations(w5,m8,1).
possible\_allocations(w5,m14,1).
%unavailability_resources(u,m,ru),
unavailability\_resources(u1,m2,1).   unavailability\_resources(u1,m3,1).
unavailability\_resources(u1,m4,1).   unavailability\_resources(u1,m5,1).
unavailability\_resources(u1,m6,1).   unavailability\_resources(u1,m7,1).
unavailability\_resources(u1,m8,1).   unavailability\_resources(u1,m9,1).
unavailability\_resources(u1,m10,1). unavailability\_resources(u1,m11,1).
unavailability\_resources(u1,m12,1). navailability\_resources(u1,m13,1).
unavailability\_resources(u1,m14,1).
%unavailability_workstations(u,w,rm),
unavailability\_workstations(u1,w1,1). unavailability\_workstations(u1,w2,1).
unavailability\_workstations(u1,w3,1). unavailability\_workstations(u1,w4,1).
unavailability\_workstations(u1,w5,1).
%sequence_constraints(o1,o2,fo),
sequence\_constraints(o1,o2,1).   sequence\_constraints(o1,o11,1).
sequence\_constraints(o1,o14,1).  sequence\_constraints(o1,o19,1).
sequence\_constraints(o2,o1,1).   sequence\_constraints(o2,o11,1).
sequence\_constraints(o2,o14,1).  sequence\_constraints(o2,o19,1).
sequence\_constraints(o3,o6,1).   sequence\_constraints(o3,o12,1).
```

```
sequence\_constraints(o3,o15,1).    sequence\_constraints(o3,o20,1).
sequence\_constraints(o4,o8,1).     sequence\_constraints(o4,o16,1).
sequence\_constraints(o5,o10,1).    sequence\_constraints(o5,o18,1).
sequence\_constraints(o6,o3,1).     sequence\_constraints(o6,o12,1).
sequence\_constraints(o6,o15,1).    sequence\_constraints(o6,o20,1).
sequence\_constraints(o7,o16,1).    sequence\_constraints(o8,o4,1).
sequence\_constraints(o8,o16,1).    sequence\_constraints(o9,o13,1).
sequence\_constraints(o9,o17,1).    sequence\_constraints(o10,o5,1).
sequence\_constraints(o10,o18,1).   sequence\_constraints(o11,o1,1).
sequence\_constraints(o11,o2,1).    sequence\_constraints(o11,o14,1).
sequence\_constraints(o11,o19,1).   sequence\_constraints(o12,o3,1).
sequence\_constraints(o12,o6,1).    sequence\_constraints(o12,o15,1).
sequence\_constraints(o12,o20,1).   sequence\_constraints(o13,o9,1).
sequence\_constraints(o13,o17,1).   sequence\_constraints(o14,o1,1).
sequence\_constraints(o14,o2,1).    sequence\_constraints(o14,o11,1).
sequence\_constraints(o14,o19,1).   sequence\_constraints(o15,o3,1).
sequence\_constraints(o15,o6,1).    sequence\_constraints(o15,o15,1).
sequence\_constraints(o15,o20,1).   sequence\_constraints(o16,o4,1).
sequence\_constraints(o16,o7,1).    sequence\_constraints(o16,o8,1).
sequence\_constraints(o17,o9,1).    sequence\_constraints(o17,o13,1).
sequence\_constraints(o18,o5,1).    sequence\_constraints(o18,o10,1).
sequence\_constraints(o19,o1,1).    sequence\_constraints(o19,o2,1).
sequence\_constraints(o19,o11,1).   sequence\_constraints(o19,o14,1).
seq
```

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
