# Peer review of "A New Approach to the Allocation of Multidimensional Resources in Production Processes"

_applsci, doi:10.3390/app12146933_

Round 1

Reviewer 1 Report

The authors present a paper entitled “A proactive and reactive approach to the allocation and control of multidimensional resources in production processes”. The paper is well structured, the research methodology, the results are correct and clearly presented. However, there are several aspect that must be improved:

-          There is an extensive literature regarding production control. Authors are encouraged to review bioinspired systems, fractal systems, and holonic manufacturing systems.

-          A review of the existing production control systems should be carried out and points for improvement identified.

-          A section must be included that identifies the improvements of the proposed model with respect to the existing ones

-          Basically, authors should enhance your findings, limitations, underscore the scientific value added of the paper.

Reviewer 2 Report

Interesting topic

The authors address the problem of allocation and control of multidimensional 19 resources in production processes.

The title of the manuscript is too long, make it concise

The abstract should list the major contributions and support it with numerical data

Instead of notations in tabulation form, prefer to list notations adopted in a list as Abbreviations and notations. This should be aligned with the format of the Journal

An introduction should list the major contributions step by step (section by section) and mention it in which sections one can be found it.

Table 4 title could be as Mathematical forms of constraints. But the question is why equations in tabulation form, why not as equations 1, 2 so on

Constraints notations in Table 4 and Table 5 are not consistent.

Need to check the consistency of notations throughout the manuscript

In Table 5 what are Q01, Q02 so on.. clarify

In section 5, support your findings and inference of the obtained results in context to the production process. Very weak analysis in section 5.

The contribution of this paper is not clear in section 7. The authors should at least compare some of the existing methodologies to show the advantage of the proposed methodology in the production process.

Future research plans should be original ones based on limitations

The conclusion should be clear and aim to leave the reader with a clear understanding of the main argument, and discussion that is presented in the manuscript.

Thank you

Round 2

Reviewer 1 Report

None

Author Response

Thank you for your review and helpful comments.

Reviewer 2 Report

Title of paper is same as earlier version

Avoid using "chapter" word in scientific manuscript

Use section instead of chapter

Need to follow journal format to represent equation, it is missing
